# A Hybrid MADM Approach for the Evaluation of Different Material Handling Issues in Flexible Manufacturing Systems

**Sandhya Dixit \*** 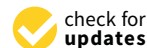 **and Tilak Raj**

Department of Mechanical Engineering, J. C. Bose University of Science and Technology, YMCA, Faridabad 121006, India; tilakraj64@rediffmail.com
* Correspondence: sandhya_parinam@yahoo.co.in

**Abstract:** In today's versatile and dynamic industrial scenario more and more industries are adopting advanced manufacturing technologies and systems like flexible manufacturing system (FMS) which combines the efficiency of a mass production line and the flexibility of a job shop. Material handling equipment form an important component of FMS and using proper material handling equipment can enhance the production process, provide effective utilization of manpower, increase production and improve system flexibility. In this research the main material handling issues in a FMS are identified and further evaluated using ISM and TOPSIS approaches. The purpose of identification of these issues and their analysis is to allow researchers and practicing managers to pay proper attention to these issues which may help them in designing the material handling systems in their organisations in a better way.

**Keywords:** manufacturing; FMS; material handling equipment; MADM; ISM; TOPSIS

## 1. Introduction

Material handling (MH) is involved at all stages from the time the raw material enters the factory till the finished product goes out. Depending on the type, a component may be handled even fifty times or more before it changes to finished product. As material handling adds no value but increases the production cycle time, it is desirable to eliminate handling wherever possible. According to Sule (1994); Sujono and Lashkari (2007), material handling accounts for 30–75% of the total cost of a product along the production chain and efficient material handling can be responsible for reducing the manufacturing system operations cost by 15–30%. It, thus, becomes clear that the cost of production of an item can be lowered considerably by making a saving in the material handling cost.

Materials handling management is among many factors that contribute to improve a company's performance (Vieira et al. 2011). Many researchers have stressed the importance of proper material handling in an organisation. Tuzkaya et al. (2010) point outs that the use of proper material handling equipment can enhance the production process and improves system flexibility. Proper selection of appropriate material handling equipment has become a most important parameter for modern manufacturing concerns (Tompkins 2010). Selecting appropriate material handling equipment can decrease manufacturing lead times, increase the efficiency of material flow, improve facility utilization and increase productivity (Kulak et al. 2004). An efficient MH system greatly improves the competitiveness of a product through the reduction of handling cost, enhances the production process, increases production and system flexibility, increases efficiency of material flow, improves facility utilization, provides effective utilization of manpower and decreases lead time (Beamon 1998).

In today's versatile and dynamic industrial scenario the importance of material handling systems is also increasing. Today, more and more industries are adopting advanced manufacturing technologies and systems to cope up with this market pressure. It has been established that the manufacturing flexibility is an important component to respond quickly to changing markets and customer demands without incurring excessive time and cost (Oberoi et al. 2008). So, industries are now opting for low volume, high variety flexible manufacturing system (FMS) instead of the conventional mass production (Dixit and Raj 2016). The mass production line efficiency and the flexibility of a job shop to produce a variety of products on a group of machines are combined in an FMS design (Rao and Parnichkun 2009). For increasing the productivity of modern manufacturing, which involves high product proliferation, FMS is very crucial and is being adopted by both large as well as small scale industries (Dai and Lee 2012).

FMS involve the general purpose manufacturing machines, coupled with material handling systems which have the capability to perform different types of operations within minimum time. In these systems, a central computer system controls the machines and material handling systems (Jahan et al. 2012). Material handling equipment form an important component of FMS and using proper material handling equipment can enhance the production process, provide effective utilization of manpower, increase production and improve system flexibility. Material handling system design is very significant in flexible manufacturing systems because most of the time that material spends on a shop floor is spent either in waiting or in transportation, even though both these activities are non-value added activities. Efficient material handling ensures for less blocking, timely delivery and reduced idle time of machines due to proper availability and gathering of materials at workstations (Ojha et al. 2018).

Groover (2001) highlights that despite its importance, materials handling is a topic that frequently is treated superficially by the companies. So, in this paper an attempt has been made to discuss the different issues related to material handling systems especially in advanced manufacturing systems like FMS. These issues are further modelled based on their importance using ISM (Interpretive Structural Modelling) approach. For the validation of the ISM model, the same issues are evaluated using TOPSIS (Technique for Order Preference by Similarity to the Ideal Solution) technique and the two results are compared and discussed.

The organization of this research paper is as follows: an introduction to different material handling issues is given in Section 2. A literature review of the techniques used in this research is presented in the Section 3. Stepwise methodologies are presented in Section 4. Modelling is done using ISM in Section 5 and TOPSIS in Section 6 respectively. Section 7 discusses the findings and the implications of this research, followed by conclusion in Section 8.

## 2. Material Handling System and Equipment

The Materials Handling Industry of America [MHIA] defines materials handling management as "Material Handling is the movement, storage, control and protection of material, goods and products throughout the process of manufacturing, distribution, consumption and disposal. The focus is on the methods, mechanical equipment, systems and related controls used to achieve these functions" (mhia.org/learning/glossary).

Material handling involves the movement of materials, manually or mechanically in batches or one item at a time within the plant. The movement may be horizontal, vertical or the combination of the both, it can be on a fixed path or on variable path. There is a need of installing safe and efficient methods and equipment for handling materials. As per MHIA, 35 to 40% of the plant accidents are due to bad methods of material handling.

According to Karande and Chakraborty (2013), the main functions performed by MH equipment can be classified into four broad categories, that is: (a) transport; (b) positioning; (c) unit formation; and (d) storage. Usually, all the MH functions are one or more combinations of these four primary functions. The transport equipments simply move materials from one point to another and mainly include conveyors, industrial trucks, cranes and so forth. Unlike transport equipment, positioning

equipment are usually employed at workstations to aid machining operations like, robots, index tables, rotary tables and so forth. Unit formation equipment are used for holding or carrying materials in standardized unit load forms for transport and storage and generally includes bins, pallets, skidsand containers. Storage equipments are used for holding or buffering materials over a period of time. Typical examples that perform this function are AS/RS, pallet racks and shelves. Today a wide variety of equipments are available for all these functions, each having distinct characteristics and cost. Selection of the proper equipment for a designed manufacturing system is a very complicated task and is often influenced by the ongoing development of new technology, practices and equipment.

As material handling adds no value but increases the production cycle time, so it is desirable to eliminate handling wherever possible. Ideally there should not be any handling at all! So, while designing the MH system, sequence the operations in logical manner so that handling is unidirectional and smooth. Wherever possible, the use of gravity is also desirable. Standardize the handling equipment to the extent possible as it means interchangeable usage, better utilization of handling equipment and lesser spares holding. In selection of handling equipment, criteria of versatility and adaptability must be the governing factor. Weight of unit load must be maximum so that each 'handling trip' is productive. Work study aspects, such as elimination of unnecessary movements and combination of processes should be considered while installing a material handling system. Application of OR techniques such as queuing can be very effective in optimal utilization of materials handling equipment. A very important aspect in the design of a material handling system is the safety aspect. The system designed should be simple and safe to operate.

The main engineering factors to be considered for the design of an efficient MH system are like: the existing plant layout and the handling equipment, the nature of the products to be handled, production processes and equipment, quantities involved. The economic factors to be considered are: The cost of material handling equipment, operating costs, repair and maintenance costs and so forth. A material handling system with the lowest prospective cost is selected. The operating costs are reduced by purchasing flexible material handling systems, increasing the amount of material to be handled at one time, minimizing the idle time for the equipment, increasing speed of handling. A material handling system is said to be economical if the cost of handling per unit weight of the material for a particular movement is minimum.

So, during the design of a materials handling system we are looking for serving equipment for a complex materials handling task and synchronizing their operations (Telek 2013). The MH equipment selection is an important proportion of manufacturing expenses and the most critical material handling decisions in this area are the arrangement and the design of material flow patterns (Asef-Vaziri and Laporte 2005). This idea is shared by Ioannou (2007), which argues that an important aspect of any production system is the design of a material handling system which integrates the production operations.

So, the decision makers have to consider various quantitative (load capacity, energy consumption, reliability, cost, etc.) and qualitative (flexibility, performance, environmental hazard, safety, load shape, load type, etc.) criteria for the design of MH systems. Some of these selection criteria are beneficial (higher values are preferred) and some are non beneficial (lower values are desired). Therefore, MH system design can be viewed as a multicriteria decision-making (MCDM) problem in the presence of many conflicting criteria (Karande and Chakraborty 2013).

Based on the above discussion, 19 issues related to the material handling equipment in FMS were identified. These issues are presented in Table 1.

**Table 1.** Major issues related to the material handling equipment in FMS.

| S. No. | Issue |
|:---:|:---:|
| 1. | Initial cost of the MH equipment |
| 2. | Load carrying capacity |
| 3. | Programming flexibility of MH equipment |
| 4. | Operational cost |
| 5. | Throughput rate |
| 6. | Capacity to handle different shapes and volumes |
| 7. | Storage/Retrieval MH equipment |
| 8. | Operational control |
| 9. | Automation |
| 10 | Floor space |
| 11. | AGVs/Robots and other advanced MH equipment already present |
| 12. | Number of AGVs required |
| 13. | Layout of AGV tracks |
| 14. | Vehicle dispatching rules |
| 15. | Traffic management |
| 16. | Positioning of idle vehicles |
| 17. | Failure management |
| 18. | Compatibility of MH equipment with other workstations |
| 19. | Comparison with cheap human labour |

## 3. Literature Review

In this section a review of the literature regarding the methodologies used in this paper is given.

### 3.1. Review of Literature Regarding ISM

Interpretive Structural Modelling is a method of establishing a relationship between the various attributes of any problem or issue. Warfield (1974) first proposed this technique and later it was used by a number of researchers to develop a visualised structure for the different elements of any complex system. ISM establishes a hierarchical relationship between different directly or indirectly related items which define a problem or an issue (Sage 1977). In this technique, both graphics as well as words are used in a carefully designed pattern to form a systematic model (Singh et al. 2003; Raj et al. 2012). The developed model helps in decision making and hence ISM is a decision making tool (Jharkharia and Shankar 2005). The ISM process transforms unclear, poorly articulated mental models of the systems into visible and well defined models (Dixit and Raj 2016).

In the past, Ravi and Shankar (2005); Faisal (2010); Luthra et al. (2011); Sharma and Bhat (2014) have applied ISM to model the various attributes of the supply chains, Raj et al. (2009) and Dixit and Raj (2016) have used it to structure FMS variables, Govindan et al. (2012) used ISM in the analysis of third party reverse logistics provider. Attri et al. (2012) and Poduval et al. (2015) have modelled the TPM environment using ISM. The drivers of agile manufacturing are modelled by Mishra et al. (2012) and Upadhye et al. (2014) studied the implementation enablers for JIT in Indian packaging industry. Analysis of sustainable manufacturing factors in Indian automotive component sector was done with ISM by Thirupathi and Vinodh (2016). The feasibility analysis of FMS in SMEs in India is done using ISM by Dixit and Raj (2018).

### 3.2. Review of Literature Regarding TOPSIS

TOPSIS is an acronym for Technique for Order Preference by Similarity to the Ideal Solution. It was developed by Hwang and Yoon (1981) and later used by many researchers like, Lai et al. (1994) and Yoon and Hwang (1995). TOPSIS is a multiple criteria decision making technique which identifies solutions from a given set of attributes based on simultaneous minimization of distance from an ideal point and maximization of distance from a nadir point (Rao 2007). Relative weights of criterion importance can be incorporated in TOPSIS.

Some of the applications in which TOPSIS have been applied in the past are: Agrawal et al. (1992) have used TOPSIS for selection of grippers in FMS, Kim et al. (1997) studied investment opportunities for AMSs, Robot selection was done by Parkan and Wu (1999), Deng et al. (2000) have used TOPSIS in

Comparing company performances, Hao and Xie (2006) developed a bidding model for evaluation of manufacturing enterprises. Wang and Chang (2007) used it for the evaluation of initial training aircraft. Kannan et al. (2009) selected reverse logistics provider and Yang and Sun (2010) used it as a new personalized recommendation technique. Personnel selection in multi-type information environment was done by Chen et al. (2011) using TOPSIS while Latpate (2015) solved a supplier selection problem in SCM. Jain and Raj (2015) modelled flexibility in FMs using TOPSIS and Ziaei et al. (2016) improved the performance of water pump manufacturing system.

## 4. Methodology

### 4.1. Interpretive Structural Modelling

The methodology for ISM modelling can be described in the following steps (Raj et al. 2008; Dixit and Raj 2018):

**Step 1:** First of all different attributes related to any issue are identified either by a survey or any group problem solving technique.

**Step 2:** Further, these different attributes are related by contextual relationship. For this a Structural self-interaction matrix (SSIM) is developed, which indicates the pair-wise relationship between the different attributes.

**Step 3:** Reachability Matrix (RM) is developed from the SSIM. The transitivity of this matrix is also checked and Final Reachability Matrix containing all the transitive links is developed.

**Step 4:** Different level partitioning is done of RM. The reachability set and antecedent set for each issue are found from the final reachability matrix. Then the intersection set is found and the levels are decided based on all the issues which can reach the issue from bottom.

**Step 5:** Conical matrix is developed with zero elements in the upper diagonal and the unitary elements in the lower half.

**Step 6:** Based on the conical matrix relationships, a directed graph (digraph) without the transitive links is drawn.

**Step 7:** The nodes of the digraph are replaced with statements to convert it into an ISM model.

### 4.2. TOPSIS

The TOPSIS methodology in a stepwise fashion for the selection of the most desirable attribute from given attributes is described below (Rao 2007):

**Step 1:** Determine the related attributes for the given objective.

**Step 2:** All the information related to the attributes is expressed in matrix form. This information may be collected from a survey or group discussion. Each row of this matrix is allocated to one attribute and each column to one criterion. Raw measurements are standardized by using Equation (1), where, raw measure $X_{ij}$ is standardized into $S_{ij}$,

$$S_{ij} = X_{ij} / \left( \left( \sum_{i=1,j=1}^{n,k} \left( X_{ij}{}^2 \right) \right)^{1/2} \right) \tag{1}$$

**Step 3:** Importance weights $w_k$ is developed for each of the criteria. The relative importance of these criteria is reflected by these weights. In this work the weightage of rating is calculated by using following criteria:

$$Normalised\ weight\ of\ each\ importamce = \frac{Total\ of\ each\ importance}{Grand\ total\ of\ all\ importances} \tag{2}$$

**Step 4:** The weighted normalized matrix $W_{ij}$ is obtained by multiplying each element of the column of the matrix $S_{ij}$ with its associated weight $w_k$. So, the elements of this are expressed as:

$$W_{ij} = w_k S_{ij} \tag{3}$$

**Step 5:** Identify the ideal attribute on each criterion, $S^+$.

**Step 6:** Identify the nadir attribute on each criterion, $S^-$.

**Step 7:** A distance measure is developed over each criterion to both ideal ($D^+$) and nadir ($D^-$).

$$D_t^+ = \left\{ \sum_{j=1}^{k} (W_{ij}^- S_j^+)^2 \right\}^{1/2} \quad i = 1,\, 2,\, 3 \ldots \ldots n \tag{4}$$

$$D_t^- = \left\{ \sum_{j=1}^{k} (W_{ij}^- S_j^-)^2 \right\}^{1/2} \quad i = 1,\, 2,\, 3 \ldots \ldots n \tag{5}$$

**Step 8:** The relative closeness of a particular attribute to the ideal solution, $R_i$, is calculated as shown in Equation (6)

$$R_i = \frac{D_i^-}{D_i^- + D_i^+} \quad i = 1,\, 2,\, 3 \ldots .. n \tag{6}$$

**Step 9:** Finally, all the alternatives are arranged in the descending order according to the value of $R_i$ indicating the most preferred and least preferred attribute.

## 5. ISM Model for the Evaluation of Material Handling Issues

In this section the ISM model is developed for the evaluation of material handling issues in FMS.

### 5.1. Development of SSIM

The data about the various issues for the ISM model is obtained by using the results of an industrial survey carried out in various industries in India. The opinion of various experts is used to form the initial conceptual relationship among the 19 issues identified in Section 2. Any conceptual inconsistencies were removed. The four symbols are used to describe the direction of relationship between any two issues, *i* and *j* have the same meaning as is used by the researchers (Dixit and Raj 2018) in their work:

Based on the contextual relationship between the issues a Structural Self-Interactive Matrix is developed for the various issues as shown in Table 2.

**Table 2.** Structural self-interactive matrix.

| Issue | 19 | 18 | 17 | 16 | 15 | 14 | 13 | 12 | 11 | 10 | 9 | 8 | 7 | 6 | 5 | 4 | 3 | 2 |
|---|---|---|---|---|---|---|---|---|---|---|---|---|---|---|---|---|---|---|
| 1 | X | O | O | O | O | O | O | A | O | O | A | O | O | A | O | O | A | A |
| 2 | V | O | O | O | V | V | V | V | A | O | O | O | O | A | V | V | O | |
| 3 | O | O | O | O | V | O | O | V | O | O | V | O | A | V | V | | | |
| 4 | V | O | O | O | O | O | O | O | O | O | O | A | O | A | O | | | |
| 5 | O | O | O | O | V | V | A | O | O | O | O | O | O | O | | | | |
| 6 | O | V | O | O | O | V | O | V | O | O | O | O | A | | | | | |
| 7 | A | V | O | V | O | O | V | O | O | O | V | O | | | | | | |
| 8 | O | V | V | O | V | V | O | O | O | O | A | | | | | | | |
| 9 | A | V | V | O | O | O | O | O | O | O | | | | | | | | |
| 10 | O | O | O | V | V | O | V | V | O | | | | | | | | | |
| 11 | O | O | V | O | O | V | V | V | | | | | | | | | | |
| 12 | O | A | O | O | V | V | V | | | | | | | | | | | |
| 13 | O | A | O | O | V | X | | | | | | | | | | | | |
| 14 | O | A | O | O | V | | | | | | | | | | | | | |
| 15 | O | O | O | V | | | | | | | | | | | | | | |
| 16 | O | O | O | | | | | | | | | | | | | | | |
| 17 | O | O | | | | | | | | | | | | | | | | |
| 18 | O | | | | | | | | | | | | | | | | | |

### 5.2. Development of the Reachability Matrix

Binary digits 0 and 1 are used in place of symbols V, A, X, O of SSIM to develop the reachability matrix. The substitution rules are same as Dixit and Raj (2018).

The initial reachability matrix is prepared and is shown in Table 3.

**Table 3.** Initial Reachability Matrix.

| Issue | 1 | 2 | 3 | 4 | 5 | 6 | 7 | 8 | 9 | 10 | 11 | 12 | 13 | 14 | 15 | 16 | 17 | 18 | 19 |
|---|---|---|---|---|---|---|---|---|---|---|---|---|---|---|---|---|---|---|---|
| 1 | 1 | 0 | 0 | 0 | 0 | 0 | 0 | 0 | 0 | 0 | 0 | 0 | 0 | 0 | 0 | 0 | 0 | 0 | 1 |
| 2 | 1 | 1 | 0 | 1 | 1 | 0 | 0 | 0 | 0 | 0 | 0 | 1 | 1 | 1 | 1 | 0 | 0 | 0 | 1 |
| 3 | 1 | 0 | 1 | 1 | 1 | 0 | 0 | 1 | 1 | 0 | 0 | 1 | 0 | 0 | 1 | 0 | 0 | 0 | 0 |
| 4 | 0 | 0 | 0 | 1 | 0 | 0 | 0 | 0 | 0 | 0 | 0 | 0 | 0 | 0 | 0 | 0 | 0 | 0 | 1 |
| 5 | 0 | 0 | 0 | 0 | 1 | 0 | 0 | 0 | 0 | 0 | 0 | 0 | 0 | 1 | 1 | 0 | 0 | 0 | 0 |
| 6 | 1 | 1 | 1 | 1 | 0 | 1 | 0 | 0 | 0 | 0 | 0 | 1 | 0 | 1 | 0 | 0 | 0 | 1 | 0 |
| 7 | 0 | 0 | 0 | 0 | 0 | 1 | 1 | 0 | 1 | 0 | 0 | 0 | 1 | 0 | 0 | 1 | 0 | 1 | 0 |
| 8 | 0 | 0 | 0 | 1 | 0 | 0 | 0 | 1 | 0 | 0 | 0 | 0 | 0 | 1 | 1 | 0 | 1 | 1 | 0 |
| 9 | 1 | 0 | 0 | 0 | 0 | 0 | 0 | 1 | 1 | 0 | 0 | 0 | 0 | 0 | 0 | 0 | 1 | 1 | 0 |
| 10 | 0 | 0 | 0 | 0 | 0 | 0 | 0 | 0 | 0 | 1 | 0 | 1 | 1 | 0 | 1 | 1 | 0 | 0 | 0 |
| 11 | 0 | 1 | 0 | 0 | 0 | 0 | 0 | 0 | 0 | 0 | 1 | 1 | 1 | 1 | 1 | 0 | 1 | 0 | 0 |
| 12 | 1 | 0 | 0 | 0 | 0 | 0 | 0 | 0 | 0 | 0 | 0 | 1 | 1 | 1 | 1 | 0 | 0 | 0 | 0 |
| 13 | 0 | 0 | 0 | 0 | 1 | 0 | 0 | 0 | 0 | 0 | 0 | 0 | 1 | 1 | 1 | 0 | 0 | 0 | 0 |
| 14 | 0 | 0 | 0 | 0 | 0 | 0 | 0 | 0 | 0 | 0 | 0 | 0 | 1 | 1 | 1 | 0 | 0 | 0 | 0 |
| 15 | 0 | 0 | 0 | 0 | 0 | 0 | 0 | 0 | 0 | 0 | 0 | 0 | 0 | 0 | 1 | 1 | 0 | 0 | 0 |
| 16 | 0 | 0 | 0 | 0 | 0 | 0 | 0 | 0 | 0 | 0 | 0 | 0 | 0 | 0 | 0 | 1 | 0 | 0 | 0 |
| 17 | 0 | 0 | 0 | 0 | 0 | 0 | 0 | 0 | 0 | 0 | 0 | 0 | 0 | 0 | 0 | 0 | 1 | 0 | 0 |
| 18 | 0 | 0 | 0 | 0 | 0 | 0 | 0 | 0 | 0 | 0 | 0 | 1 | 1 | 1 | 0 | 0 | 0 | 1 | 0 |
| 19 | 1 | 0 | 0 | 0 | 0 | 0 | 1 | 0 | 1 | 0 | 0 | 0 | 0 | 0 | 0 | 0 | 0 | 0 | 1 |

The final reachability matrix is obtained from the initial matrix by incorporating transitivity that is, if an issue *a* influences issue *b* and issue *b* further influences issue *c*, the issue *a* automatically influences issue *c*. The final reachability matrix obtained by incorporating transitivity is shown in Table 4 wherein transitive influences are shown as 1 *.

**Table 4.** Final Reachability Matrix.

| Issue | 1 | 2 | 3 | 4 | 5 | 6 | 7 | 8 | 9 | 10 | 11 | 12 | 13 | 14 | 15 | 16 | 17 | 18 | 19 |
|---|---|---|---|---|---|---|---|---|---|---|---|---|---|---|---|---|---|---|---|
| 1 | 1 | 0 | 0 | 0 | 0 | 1 * | 1 * | 1 * | 1 * | 0 | 0 | 0 | 1 * | 0 | 0 | 1 * | 1 * | 1 * | 1 |
| 2 | 1 | 1 | 0 | 1 | 1 | 1 * | 1 * | 1 * | 1 * | 0 | 0 | 1 | 1 | 1 | 1 | 1 * | 1 * | 1 * | 1 |
| 3 | 1 | 0 | 1 | 1 | 1 | 0 | 1 * | 1 | 1 | 0 | 0 | 1 | 1 * | 1 * | 1 | 1 * | 1 * | 1 * | 1 * |
| 4 | 1 * | 0 | 0 | 1 | 0 | 1 * | 1 * | 1 * | 1 * | 0 | 0 | 0 | 1 * | 0 | 0 | 1 * | 1 * | 1 * | 1 |
| 5 | 0 | 0 | 0 | 0 | 1 | 0 | 0 | 0 | 0 | 0 | 0 | 0 | 1 * | 1 | 1 | 1 * | 0 | 0 | 0 |
| 6 | 1 | 1 | 1 | 1 | 1 * | 1 | 0 | 1 * | 1 * | 0 | 0 | 1 | 1 * | 1 | 1 * | 0 | 1 * | 1 | 1 * |
| 7 | 1 * | 1 * | 1 * | 1 * | 1 * | 1 | 1 | 1 * | 1 | 0 | 0 | 1 * | 1 | 1 * | 1 * | 1 | 1 * | 1 | 0 |
| 8 | 1 * | 0 | 0 | 1 | 1 * | 0 | 1 * | 1 | 1 * | 0 | 0 | 1 * | 1 * | 1 | 1 | 1 * | 1 | 1 | 1 * |
| 9 | 1 | 0 | 0 | 1 * | 1 * | 0 | 1 * | 1 | 1 | 0 | 0 | 1 * | 1 * | 1 * | 1 * | 1 * | 1 | 1 | 1 * |
| 10 | 1 * | 0 | 0 | 0 | 1 * | 0 | 0 | 0 | 0 | 1 | 0 | 1 | 1 | 1 * | 1 | 1 | 0 | 0 | 1 * |
| 11 | 1 * | 1 | 0 | 1 * | 1 * | 0 | 1 * | 0 | 1 * | 0 | 1 | 1 | 1 | 1 | 1 * | 1 * | 1 | 0 | 1 * |
| 12 | 1 | 0 | 0 | 0 | 1 * | 0 | 1 * | 0 | 1 * | 0 | 0 | 1 | 1 | 1 | 1 | 1 * | 0 | 0 | 1 * |
| 13 | 0 | 0 | 0 | 0 | 1 | 0 | 0 | 0 | 0 | 0 | 0 | 0 | 1 | 1 | 1 | 1 * | 0 | 0 | 0 |
| 14 | 0 | 0 | 0 | 0 | 1 * | 0 | 0 | 0 | 0 | 0 | 0 | 0 | 1 | 1 | 1 | 1 * | 0 | 0 | 0 |
| 15 | 0 | 0 | 0 | 0 | 0 | 0 | 0 | 0 | 0 | 0 | 0 | 0 | 0 | 0 | 1 | 1 | 0 | 0 | 0 |
| 16 | 0 | 0 | 0 | 0 | 0 | 0 | 0 | 0 | 0 | 0 | 0 | 0 | 0 | 0 | 0 | 1 | 0 | 0 | 0 |
| 17 | 0 | 0 | 0 | 0 | 0 | 0 | 0 | 0 | 0 | 0 | 0 | 0 | 0 | 0 | 0 | 0 | 1 | 0 | 0 |
| 18 | 1 * | 0 | 0 | 0 | 1 * | 0 | 0 | 0 | 0 | 0 | 0 | 1 | 1 | 1 | 1 * | 0 | 0 | 1 | 1 * |
| 19 | 1 | 1 * | 1 * | 1 * | 0 | 1 * | 1 | 1 * | 1 | 0 | 0 | 1 * | 1 * | 1 * | 1 * | 1 * | 1 * | 1 * | 1 |

### 5.3. Partitioning the Reachability Matrix

The reachability and the antecedent sets for each issue are found by partitioning the final reachability matrix is (Warfield 1974). The process is completed in twelve iterations giving twelve different levels for the ISM model. These levels are shown in Tables 5–16.

**Table 5.** Iteration 1.

| Issue | Reachability Set | Antecedent Set | Intersection Set | Level |
|---|---|---|---|---|
| 1 | 1,6,7,8,9,13,16,17,18,19 | 1,2,3,4,6,7,8,9,10,11,12,18,19 | 1,6,7,8,9,18,19 | |
| 2 | 1,2,4,5,6,7,8,9,12,13,14,15,16,17,18,19 | 2,6,7,11,19 | 2,6,7,19 | |
| 3 | 1,3,4,5,7,8,9,12,13,14,15,16,17,18,19 | 3,6,7,19 | 3,7,19 | |
| 4 | 1,4,6,7,8,9,13,16,17,18,19 | 2,3,4,6,7,8,9,11,19 | 4,6,7,8,9,19 | |
| 5 | 5, 13,14,15,16 | 1,2,5,6,7,8,9,10,11,12,13,14,18 | 5,13,14 | |
| 6 | 1,2,3,4,5,6,8,9,12,13,14,15,17,18,19 | 1,2,4,6,7,19 | 1,2,4,6,19 | |
| 7 | 1,2,3,4,5,6,7,8,9,12,13,14,15,16,17,18 | 1,2,3,4,7,8,9,11,12,19 | 1,2,3,4,7,8,9,12 | |
| 8 | 1,4,5,7,8,9,12,13,14,15,16,17,18,19 | 1,2,3,4,6,7,8,9,19 | 1,4,7,8,9,19 | |
| 9 | 1,4,5,7,8,9,12,13,14,15,16,17,18,19 | 1,2,3,4,6,7,8,9,11,12,19 | 1,4,7,8,9,12,19 | |
| 10 | 1,5,10,12,13,14,15,16,19 | 10 | 10 | |
| 11 | 1,2,4,5,7,9,11,12,13,14,15,16,17,19 | 11 | 11 | |
| 12 | 1,5,7,9,12,13,14,15,16,19 | 2,3,6,7,8,9,10,11,12,18,19 | 7,9,12,19 | |
| 13 | 5,13,14,15,16 | 1,2,3,4,5,6,7,8,9,10,11,12,13,14,18,19 | 5,13,14 | |
| 14 | 5,13,14,15,16 | 2,3,5,6,7,8,9,10,11,12,13,14,18,19 | 5,13,14 | |
| 15 | 15,16 | 2,3,5,6,7,8,9,10,11,12,13,14,15,18,19 | 15 | |
| 16 | 16 | 1,2,3,4,5,7,8,9,10,11,12,13,14,15,16,19 | 16 | I |
| 17 | 17 | 1,2,3,4,6,7,8,9,11,17,19 | 17 | I |
| 18 | 1,5,12,13,14,15,18,19 | 1,2,3,4,6,7,8,9,18,19 | 1,18,19 | |
| 19 | 1,2,3,4,6,7,8,9,12,13,14,15,16,17,18,19 | 1,2,3,4,6,8,9,10,11,12,18,19 | 1,2,3,4,6,8,9,12,18,19 | |

**Table 6.** Iteration 2.

| Issue | Reachability Set | Antecedent Set | Intersection Set | Level |
|---|---|---|---|---|
| 1 | 1,6,7,8,9,13,18,19 | 1,2,3,4,6,7,8,9,10,11,12,18,19 | 1,6,7,8,9,18,19 | |
| 2 | 1,2,4,5,6,7,8,9,12,13,14,15,18,19 | 2,6,7,11,19 | 2,6,7,19 | |
| 3 | 1,3,4,5,7,8,9,12,13,14,15,18,19 | 3,6,7,19 | 3,7,19 | |
| 4 | 1,4,6,7,8,9,13,18,19 | 2,3,4,6,7,8,9,11,19 | 4,6,7,8,9,19 | |
| 5 | 5, 13,14,15 | 1,2,5,6,7,8,9,10,11,12,13,14,18 | 5,13,14 | |
| 6 | 1,2,3,4,5,6,8,9,12,13,14,15,18,19 | 1,2,4,6,7,19 | 1,2,4,6,19 | |
| 7 | 1,2,3,4,5,6,7,8,9,12,13,14,15,18 | 1,2,3,4,7,8,9,11,12,19 | 1,2,3,4,7,8,9,12 | |
| 8 | 1,4,5,7,8,9,12,13,14,15,18,19 | 1,2,3,4,6,7,8,9,19 | 1,4,7,8,9,19 | |
| 9 | 1,4,5,7,8,9,12,13,14,15,18,19 | 1,2,3,4,6,7,8,9,11,12,19 | 1,4,7,8,9,12,19 | |
| 10 | 1,5,10,12,13,14,15,19 | 10 | 10 | |
| 11 | 1,2,4,5,7,9,11,12,13,14,15,19 | 11 | 11 | |
| 12 | 1,5,7,9,12,13,14,15,19 | 2,3,6,7,8,9,10,11,12,18,19 | 7,9,12,19 | |
| 13 | 5,13,14,15 | 1,2,3,4,5,6,7,8,9,10,11,12,13,14,18,19 | 5,13,14 | |
| 14 | 5,13,14,15 | 2,3,5,6,7,8,9,10,11,12,13,14,18,19 | 5,13,14 | |
| 15 | 15 | 2,3,5,6,7,8,9,10,11,12,13,14,15,18,19 | 15 | II |
| 18 | 1,5,12,13,14,15,18,19 | 1,2,3,4,6,7,8,9,18,19 | 1,18,19 | |
| 19 | 1,2,3,4,6,7,8,9,12,13,14,15,18,19 | 1,2,3,4,6,8,9,10,11,12,18,19 | 1,2,3,4,6,8,9,12,18,19 | |

**Table 7.** Iteration 3.

| Issue | Reachability Set | Antecedent Set | Intersection Set | Level |
|---|---|---|---|---|
| 1 | 1,6,7,8,9,13,18,19 | 1,2,3,4,6,7,8,9,10,11,12,18,19 | 1,6,7,8,9,18,19 | |
| 2 | 1,2,4,5,6,7,8,9,12,13,14,18,19 | 2,6,7,11,19 | 2,6,7,19 | |
| 3 | 1,3,4,5,7,8,9,12,13,14,18,19 | 3,6,7,19 | 3,7,19 | |
| 4 | 1,4,6,7,8,9,13,18,19 | 2,3,4,6,7,8,9,11,19 | 4,6,7,8,9,19 | |
| 5 | 5, 13,14 | 1,2,5,6,7,8,9,10,11,12,13,14,18 | 5,13,14 | III |
| 6 | 1,2,3,4,5,6,8,9,12,13,14,18,19 | 1,2,4,6,7,19 | 1,2,4,6,19 | |
| 7 | 1,2,3,4,5,6,7,8,9,12,13,14,18 | 1,2,3,4,7,8,9,11,12,19 | 1,2,3,4,7,8,9,12 | |
| 8 | 1,4,5,7,8,9,12,13,14,18,19 | 1,2,3,4,6,7,8,9,19 | 1,4,7,8,9,19 | |
| 9 | 1,4,5,7,8,9,12,13,14,18,19 | 1,2,3,4,6,7,8,9,11,12,19 | 1,4,7,8,9,12,19 | |
| 10 | 1,5,10,12,13,14,19 | 10 | 10 | |
| 11 | 1,2,4,5,7,9,11,12,13,14,19 | 11 | 11 | |
| 12 | 1,5,7,9,12,13,14,19 | 2,3,6,7,8,9,10,11,12,18,19 | 7,9,12,19 | |
| 13 | 5,13,14 | 1,2,3,4,5,6,7,8,9,10,11,12,13,14,18,19 | 5,13,14 | III |
| 14 | 5,13,14 | 2,3,5,6,7,8,9,10,11,12,13,14,18,19 | 5,13,14 | III |
| 18 | 1,5,12,13,14,18,19 | 1,2,3,4,6,7,8,9,18,19 | 1,18,19 | |
| 19 | 1,2,3,4,6,7,8,9,12,13,14,18,19 | 1,2,3,4,6,8,9,10,11,12,18,19 | 1,2,3,4,6,8,9,12,18,19 | |

**Table 8.** Iteration 4.

| Issue | Reachability Set | Antecedent Set | Intersection Set | Level |
|---|---|---|---|---|
| 1 | 1,6,7,8,9,18,19 | 1,2,3,4,6,7,8,9,10,11,12,18,19 | 1,6,7,8,9,18,19 | **IV** |
| 2 | 1,2,4,6,7,8,9,12,18,19 | 2,6,7,11,19 | 2,6,7,19 | |
| 3 | 1,3,4,7,8,9,12,18,19 | 3,6,7,19 | 3,7,19 | |
| 4 | 1,4,6,7,8,9,18,19 | 2,3,4,6,7,8,9,11,19 | 4,6,7,8,9,19 | |
| 6 | 1,2,3,4,6,8,9,12,18,19 | 1,2,4,6,7,19 | 1,2,4,6,19 | |
| 7 | 1,2,3,4,6,7,8,9,12,18 | 1,2,3,4,7,8,9,11,12,19 | 1,2,3,4,7,8,9,12 | |
| 8 | 1,4,7,8,9,12,18,19 | 1,2,3,4,6,7,8,9,19 | 1,4,7,8,9,19 | |
| 9 | 1,4,7,8,9,12,18,19 | 1,2,3,4,6,7,8,9,11,12,19 | 1,4,7,8,9,12,19 | |
| 10 | 1,10,12,19 | 10 | 10 | |
| 11 | 1,2,4,7,9,11,12,19 | 11 | 11 | |
| 12 | 1,7,9,12,19 | 2,3,6,7,8,9,10,11,12,18,19 | 7,9,12,19 | |
| 18 | 1,12,18,19 | 1,2,3,4,6,7,8,9,18,19 | 1,18,19 | |
| 19 | 1,2,3,4,6,7,8,9,12,18,19 | 1,2,3,4,6,8,9,10,11,12,18,19 | 1,2,3,4,6,8,9,12,18,19 | |

**Table 9.** Iteration 5.

| Issue | Reachability Set | Antecedent Set | Intersection Set | Level |
|---|---|---|---|---|
| 2 | 2,4,6,7,8,9,12,18,19 | 2,6,7,11,19 | 2,6,7,19 | |
| 3 | 3,4,7,8,9,12,18,19 | 3,6,7,19 | 3,7,19 | |
| 4 | 4,6,7,8,9,18,19 | 2,3,4,6,7,8,9,11,19 | 4,6,7,8,9,19 | |
| 6 | 2,3,4,6,8,9,12,18,19 | 2,4,6,7,19 | 2,4,6,19 | |
| 7 | 2,3,4,6,7,8,9,12,18 | 2,3,4,7,8,9,11,12,19 | 2,3,4,7,8,9,12 | |
| 8 | 4,7,8,9,12,18,19 | 2,3,4,6,7,8,9,19 | 4,7,8,9,19 | |
| 9 | 4,7,8,9,12,18,19 | 2,3,4,6,7,8,9,11,12,19 | 4,7,8,9,12,19 | |
| 10 | 10,12,19 | 10 | 10 | |
| 11 | 2,4,7,9,11,12,19 | 11 | 11 | |
| 12 | 7,9,12,19 | 2,3,6,7,8,9,10,11,12,18,19 | 7,9,12,19 | **V** |
| 18 | 12,18,19 | 2,3,4,6,7,8,9,18,19 | 18,19 | |
| 19 | 2,3,4,6,7,8,9,12,18,19 | 2,3,4,6,8,9,10,11,12,18,19 | 2,3,4,6,8,9,12,18,19 | |

**Table 10.** Iteration 6.

| Issue | Reachability Set | Antecedent Set | Intersection Set | Level |
|---|---|---|---|---|
| 2 | 2,4,6,7,8,9,18,19 | 2,6,7,11,19 | 2,6,7,19 | |
| 3 | 3,4,7,8,9,18,19 | 3,6,7,19 | 3,7,19 | |
| 4 | 4,6,7,8,9,18,19 | 2,3,4,6,7,8,9,11,19 | 4,6,7,8,9,19 | |
| 6 | 2,3,4,6,8,9,18,19 | 2,4,6,7,19 | 2,4,6,19 | |
| 7 | 2,3,4,6,7,8,9,18 | 2,3,4,7,8,9,11,19 | 2,3,4,7,8,9 | |
| 8 | 4,7,8,9,18,19 | 2,3,4,6,7,8,9,19 | 4,7,8,9,19 | |
| 9 | 4,7,8,9,18,19 | 2,3,4,6,7,8,9,11,19 | 4,7,8,9,19 | |
| 10 | 10,19 | 10 | 10 | |
| 11 | 2,4,7,9,11,19 | 11 | 11 | |
| 18 | 18,19 | 2,3,4,6,7,8,9,18,19 | 18,19 | **VI** |
| 19 | 2,3,4,6,7,8,9,18,19 | 2,3,4,6,8,9,10,11,18,19 | 2,3,4,6,8,9,18,19 | |

**Table 11.** Iteration 7.

| Issue | Reachability Set | Antecedent Set | Intersection Set | Level |
|---|---|---|---|---|
| 2 | 2,4,6,7,8,9,19 | 2,6,7,11,19 | 2,6,7,19 | |
| 3 | 3,4,7,8,9,19 | 3,6,7,19 | 3,7,19 | |
| 4 | 4,6,7,8,9,19 | 2,3,4,6,7,8,9,11,19 | 4,6,7,8,9,19 | **VII** |
| 6 | 2,3,4,6,8,9,19 | 2,4,6,7,19 | 2,4,6,19 | |
| 7 | 2,3,4,6,7,8,9 | 2,3,4,7,8,9,11,19 | 2,3,4,7,8,9 | |
| 8 | 4,7,8,9,19 | 2,3,4,6,7,8,9,19 | 4,7,8,9,19 | **VII** |
| 9 | 4,7,8,9,19 | 2,3,4,6,7,8,9,11,19 | 4,7,8,9,19 | **VII** |
| 10 | 10,19 | 10 | 10 | |
| 11 | 2,4,7,9,11,19 | 11 | 11 | |
| 19 | 2,3,4,6,7,8,9,19 | 2,3,4,6,8,9,10,11,19 | 2,3,4,6,8,9,19 | |

**Table 12.** Iteration 8.

| Issue | Reachability Set | Antecedent Set | Intersection Set | Level |
|---|---|---|---|---|
| 2 | 2,6,7,19 | 2,6,7,11,19 | 2,6,7,19 | **VIII** |
| 3 | 3,7,19 | 3,6,7,19 | 3,7,19 | **VIII** |
| 6 | 2,3,6,19 | 2,6,7,19 | 2,6,19 | |
| 7 | 2,3,6,7 | 2,3,7,11,19 | 2,3,7 | |
| 10 | 10,19 | 10 | 10 | |
| 11 | 2,7,11,19 | 11 | 11 | |
| 19 | 2,3,6,7,19 | 2,3,6,10,11,19 | 2,3,6,19 | |

**Table 13.** Iteration 9.

| Issue | Reachability Set | Antecedent Set | Intersection Set | Level |
|---|---|---|---|---|
| 6 | 6,19 | 6,7,19 | 6,19 | **IX** |
| 7 | 6,7 | 7,11,19 | 7 | |
| 10 | 10,19 | 10 | 10 | |
| 11 | 7,11,19 | 11 | 11 | |
| 19 | 6,7,19 | 6,10,11,19 | 6,19 | |

**Table 14.** Iteration 10.

| Issue | Reachability Set | Antecedent Set | Intersection Set | Level |
|---|---|---|---|---|
| 7 | 7 | 7,11,19 | 7 | **X** |
| 10 | 10,19 | 10 | 10 | |
| 11 | 7,11,19 | 11 | 11 | |
| 19 | 7,19 | 10,11,19 | 19 | |

**Table 15.** Iteration 11.

| Issue | Reachability Set | Antecedent Set | Intersection Set | Level |
|---|---|---|---|---|
| 10 | 10,19 | 10 | 10 | |
| 11 | 11,19 | 11 | 11 | |
| 19 | 19 | 10,11,19 | 19 | **XI** |

**Table 16.** Iteration 12.

| Issue | Reachability Set | Antecedent Set | Intersection Set | Level |
|---|---|---|---|---|
| 10 | 10 | 10 | 10 | **XII** |
| 11 | 11 | 11 | 11 | **XII** |

### 5.4. Development of the Conical Matrix

In the next step, the issues in the same level are clubbed together to form a conical matrix. As shown in Table 17. The summing up of the number of ones in rows gives the drive power of an issue and the summing up of the ones across columns gives the dependence power of the issue. Next, drive power and dependence power ranks are calculated by giving highest ranks to the issues that have the maximum number of ones in the rows and columns respectively (Raj et al. 2008; Attri et al. 2013).

### 5.5. Development of the Digraph and the ISM Model

From the conical matrix the digraph (Figure 1) of nodes and lines is obtained by placing the top level issues at the top of the digraph and second level issues at second position and so on, until the bottom level is placed at the lowest position in the digraph (Raj et al. 2008; Attri et al. 2013).

**Table 17.** Conical Matrix.

| Issue | 16 | 17 | 15 | 5 | 13 | 14 | 1 | 12 | 18 | 4 | 8 | 9 | 2 | 3 | 6 | 7 | 19 | 10 | 11 | Driving Power |
|---|---|---|---|---|---|---|---|---|---|---|---|---|---|---|---|---|---|---|---|---|
| 16 | 1 | 0 | 0 | 0 | 0 | 0 | 0 | 0 | 0 | 0 | 0 | 0 | 0 | 0 | 0 | 0 | 0 | 0 | 0 | 1 |
| 17 | 0 | 1 | 0 | 0 | 0 | 0 | 0 | 0 | 0 | 0 | 0 | 0 | 0 | 0 | 0 | 0 | 0 | 0 | 0 | 1 |
| 15 | 1 | 0 | 1 | 0 | 0 | 0 | 0 | 0 | 0 | 0 | 0 | 0 | 0 | 0 | 0 | 0 | 0 | 0 | 0 | 2 |
| 5 | 1 | 0 | 1 | 1 | 1 | 1 | 0 | 0 | 0 | 0 | 0 | 0 | 0 | 0 | 0 | 0 | 0 | 0 | 0 | 5 |
| 13 | 1 | 0 | 1 | 1 | 1 | 1 | 0 | 0 | 0 | 0 | 0 | 0 | 0 | 0 | 0 | 0 | 0 | 0 | 0 | 5 |
| 14 | 1 | 0 | 1 | 1 | 1 | 1 | 0 | 0 | 0 | 0 | 0 | 0 | 0 | 0 | 0 | 0 | 0 | 0 | 0 | 5 |
| 1 | 1 | 1 | 0 | 0 | 1 | 0 | 1 | 0 | 1 | 0 | 1 | 1 | 0 | 0 | 1 | 1 | 1 | 0 | 0 | 10 |
| 12 | 1 | 0 | 1 | 1 | 1 | 1 | 1 | 1 | 0 | 0 | 0 | 1 | 0 | 0 | 0 | 1 | 1 | 0 | 0 | 10 |
| 18 | 0 | 0 | 1 | 1 | 1 | 1 | 1 | 1 | 1 | 0 | 0 | 0 | 0 | 0 | 0 | 1 | 0 | 0 | 0 | 8 |
| 4 | 1 | 1 | 0 | 0 | 1 | 0 | 1 | 0 | 1 | 1 | 1 | 1 | 0 | 0 | 1 | 1 | 1 | 0 | 0 | 11 |
| 8 | 1 | 1 | 1 | 1 | 1 | 1 | 1 | 1 | 1 | 1 | 1 | 1 | 0 | 0 | 1 | 1 | 0 | 0 | 14 |
| 9 | 1 | 1 | 1 | 1 | 1 | 1 | 1 | 1 | 1 | 1 | 1 | 1 | 0 | 0 | 0 | 1 | 1 | 0 | 0 | 14 |
| 2 | 1 | 1 | 1 | 1 | 1 | 1 | 1 | 1 | 1 | 1 | 1 | 1 | 1 | 0 | 1 | 1 | 1 | 0 | 0 | 16 |
| 3 | 1 | 1 | 1 | 1 | 1 | 1 | 1 | 1 | 1 | 1 | 1 | 1 | 0 | 1 | 0 | 1 | 1 | 0 | 0 | 15 |
| 6 | 0 | 1 | 1 | 1 | 1 | 1 | 1 | 1 | 1 | 1 | 1 | 1 | 1 | 1 | 1 | 0 | 1 | 0 | 0 | 15 |
| 7 | 1 | 1 | 1 | 1 | 1 | 1 | 1 | 1 | 1 | 1 | 1 | 1 | 1 | 1 | 1 | 1 | 0 | 0 | 0 | 16 |
| 19 | 1 | 1 | 1 | 0 | 1 | 1 | 1 | 1 | 1 | 1 | 1 | 1 | 1 | 1 | 1 | 1 | 1 | 0 | 0 | 16 |
| 10 | 1 | 0 | 1 | 1 | 1 | 1 | 1 | 1 | 0 | 0 | 0 | 0 | 0 | 0 | 0 | 0 | 1 | 1 | 0 | 9 |
| 11 | 1 | 1 | 1 | 1 | 1 | 1 | 1 | 1 | 0 | 1 | 0 | 1 | 1 | 0 | 0 | 1 | 1 | 0 | 1 | 14 |
| Dependence Power | 16 | 11 | 15 | 13 | 16 | 14 | 13 | 11 | 10 | 9 | 9 | 11 | 5 | 4 | 6 | 10 | 12 | 1 | 1 | |

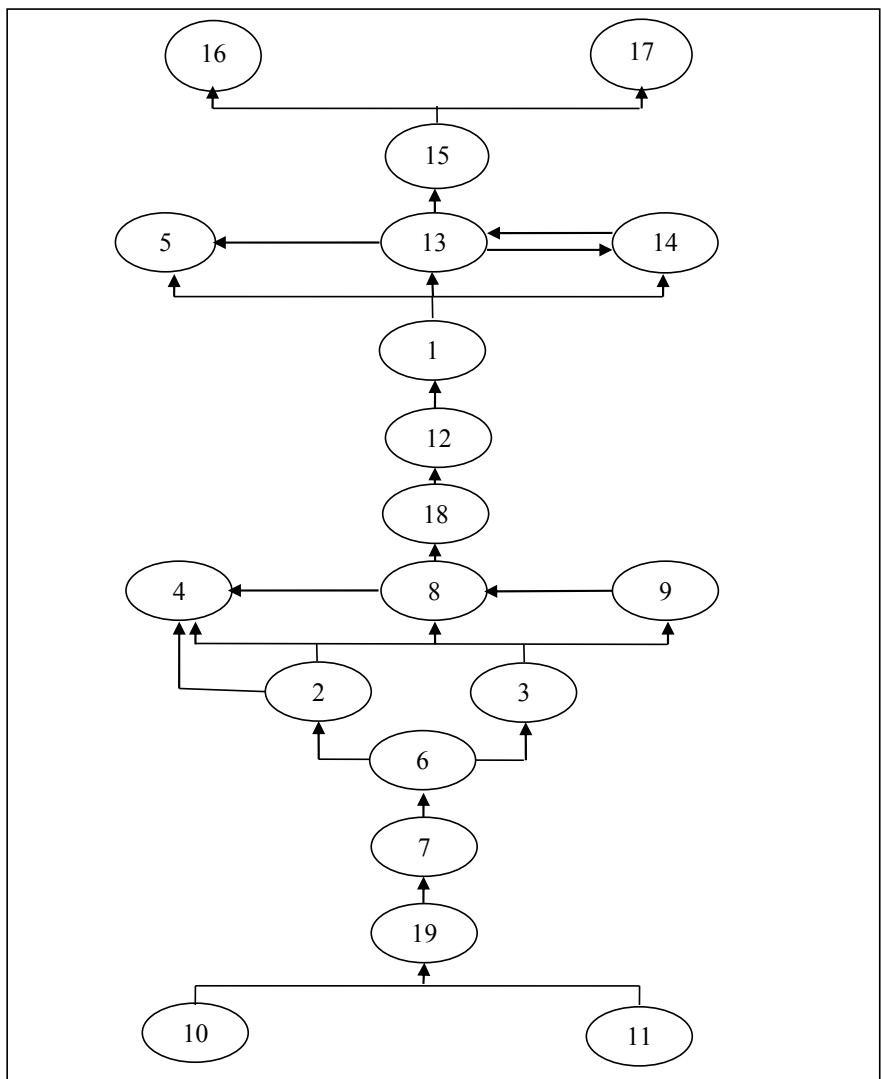

**Figure 1.** A digraph showing the relationship between different material handling issues in FMS.

Now, this digraph is converted into the actual ISM model by replacing the nodes by MH issue statements as shown in Figure 2.

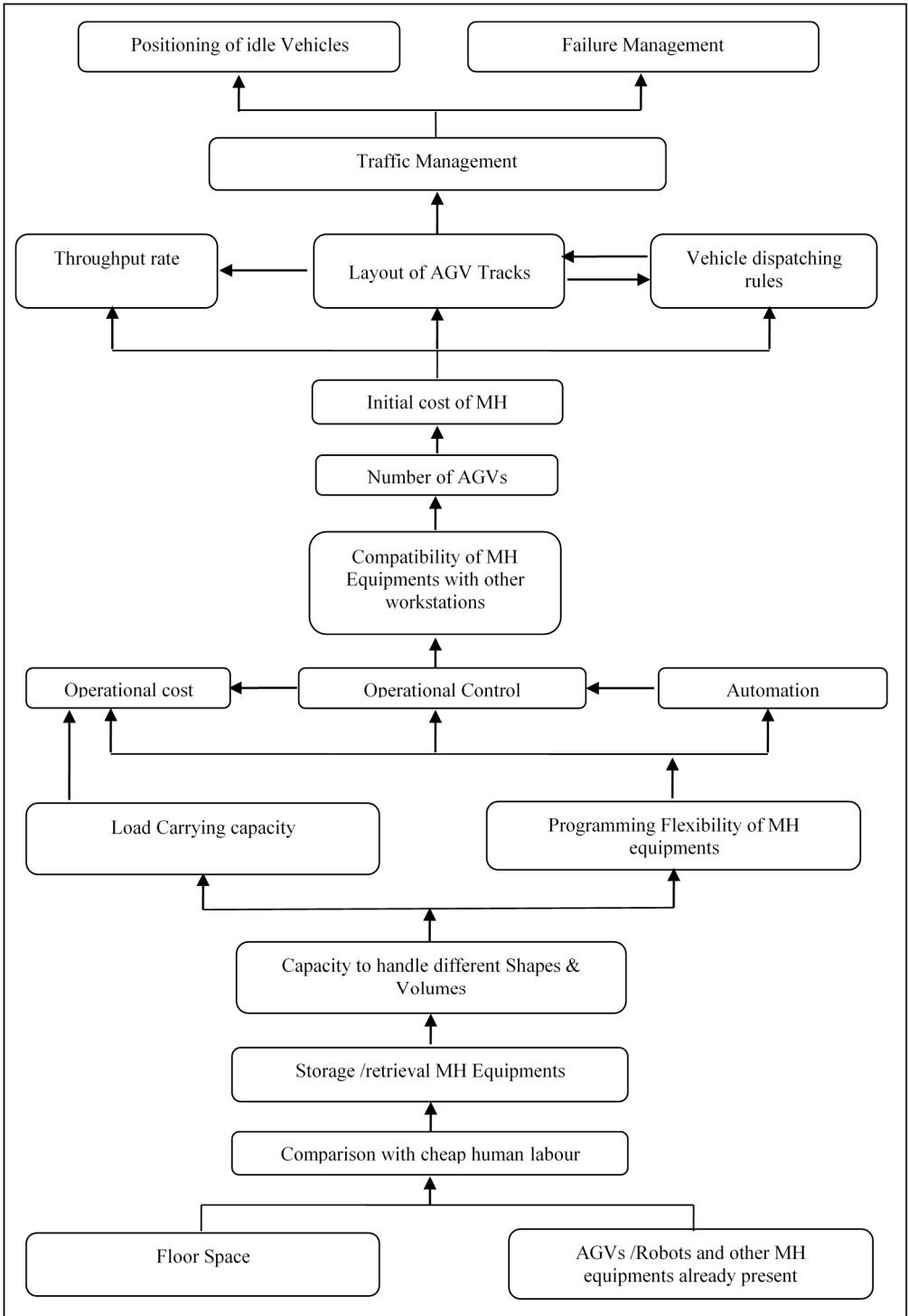

**Figure 2.** An interpretive structural model showing the levels of material handling issues.

## 6. TOPSIS Model for the Evaluation of Material Handling Issues

In this section the modelling of the MH issues for FMS is made using TOPSIS. The various steps which are described in Section 4.2 are followed as under:

**Step 1:** Objective is to find the significance hierarchy of different material handling issues. For this 19 issues were identified as given in Table 1.

**Step 2:** The next step is to represent all the information available for the issues in the form of a decision matrix. For this step, the data is collected from a survey and 19 issues of material handling in FMS were rated on a scale of 5 in which responses from 63 respondents were collected. Thus, number of attributes is, $n = 19$ and criteria, $k = 5$. This raw data in the form of frequency table is shown in Table 18.

**Table 18.** Data collected through Survey.

| Rating | 5 | 4 | 3 | 2 | 1 |
|---|---|---|---|---|---|
| Issue | Most Important | Above Average | Average | Below Average | Least Important |
| 1 | 2 | 5 | 24 | 28 | 4 |
| 2 | 1 | 10 | 33 | 15 | 4 |
| 3 | 0 | 10 | 34 | 16 | 3 |
| 4 | 0 | 23 | 28 | 12 | 0 |
| 5 | 5 | 4 | 29 | 20 | 5 |
| 6 | 12 | 9 | 34 | 8 | 0 |
| 7 | 6 | 18 | 34 | 2 | 3 |
| 8 | 2 | 28 | 15 | 18 | 0 |
| 9 | 0 | 21 | 30 | 12 | 0 |
| 10 | 4 | 9 | 50 | 0 | 0 |
| 11 | 5 | 34 | 22 | 2 | 0 |
| 12 | 0 | 11 | 30 | 22 | 0 |
| 13 | 6 | 8 | 22 | 27 | 0 |
| 14 | 7 | 14 | 30 | 12 | 0 |
| 15 | 17 | 17 | 18 | 10 | 1 |
| 16 | 2 | 18 | 13 | 24 | 6 |
| 17 | 4 | 2 | 18 | 28 | 11 |
| 18 | 11 | 25 | 25 | 2 | 0 |
| 19 | 3 | 4 | 50 | 6 | 0 |

Now the standardized matrix is obtained by normalizing the data of Table 18 using the Equation (1). The same is shown in Table 19.

**Table 19.** Data in Normalised Form.

| Rating | 5 | 4 | 3 | 2 | 1 |
|---|---|---|---|---|---|
| Issue | Most Important | Above Average | Average | Below Average | Least Important |
| 1 | 0.1433 | 0.3442 | 4.4057 | 10.8659 | 1.0482 |
| 2 | 0.0358 | 1.3767 | 8.3295 | 3.1184 | 1.0482 |
| 3 | 0.0000 | 1.3767 | 8.8420 | 3.5480 | 0.5896 |
| 4 | 0.0000 | 7.2829 | 5.9966 | 1.9958 | 0.0000 |
| 5 | 0.8957 | 0.2203 | 6.4326 | 5.5438 | 1.6378 |
| 6 | 5.1593 | 1.1151 | 8.8420 | 0.8870 | 0.0000 |
| 7 | 1.2898 | 4.4606 | 8.8420 | 0.0554 | 0.5896 |
| 8 | 0.1433 | 10.7935 | 1.7210 | 4.4905 | 0.0000 |
| 9 | 0.0000 | 6.0714 | 6.8839 | 1.9958 | 0.0000 |
| 10 | 0.5733 | 1.1151 | 19.1219 | 0.0000 | 0.0000 |
| 11 | 0.8957 | 15.9150 | 3.7020 | 0.0554 | 0.0000 |
| 12 | 0.0000 | 1.6658 | 6.8839 | 6.7080 | 0.0000 |
| 13 | 1.2898 | 0.8811 | 3.7020 | 10.1036 | 0.0000 |
| 14 | 1.7556 | 2.6984 | 6.8839 | 1.9958 | 0.0000 |
| 15 | 10.3545 | 3.9787 | 2.4782 | 1.3860 | 0.0655 |
| 16 | 0.1433 | 4.4606 | 1.2926 | 7.9831 | 2.3584 |
| 17 | 0.5733 | 0.0551 | 2.4782 | 10.8659 | 7.9270 |
| 18 | 4.3353 | 8.6045 | 4.7805 | 0.0554 | 0.0000 |
| 19 | 0.3225 | 0.2203 | 19.1219 | 0.4989 | 0.0000 |

**Step 3:** Table 20 shows these criteria weights developed for each criteria using Equation (2).

**Table 20.** Weightage of Rating.

| Rating | Most Important | Above Average | Average | Below Average | Least Important |
|---|---|---|---|---|---|
| Instance of each importance | 87 | 270 | 539 | 264 | 37 |
| Total of each importance | 435 | 1080 | 1617 | 528 | 37 |
| Normalized weight of each importance | 0.1177 | 0.2921 | 0.4374 | 0.1428 | 0.0100 |

**Step 4:** Now the weighted matrix is developed by multiplying each value of a rating column by its respective weight. The same is shown in Table 21.

**Table 21.** Weighted Matrix of Normalized Data.

| Rating | 5 | 4 | 3 | 2 | 1 |
|---|---|---|---|---|---|
| Issue | Most Important | Above Average | Average | Below Average | Least Important |
| 1 | 0.0252 | 0.1360 | 1.3506 | 1.1561 | 0.0160 |
| 2 | 0.0063 | 0.5441 | 2.5576 | 0.3318 | 0.0160 |
| 3 | 0.0000 | 0.5441 | 2.7152 | 0.3775 | 0.0090 |
| 4 | 0.0000 | 2.8782 | 1.8400 | 0.2123 | 0.0000 |
| 5 | 0.1572 | 0.0871 | 1.9741 | 0.5899 | 0.0251 |
| 6 | 0.9055 | 0.4407 | 2.7152 | 0.0944 | 0.0000 |
| 7 | 0.2264 | 1.7628 | 2.7152 | 0.0059 | 0.0090 |
| 8 | 0.0252 | 4.2656 | 0.5248 | 0.4778 | 0.0000 |
| 9 | 0.0000 | 2.3994 | 2.1129 | 0.2123 | 0.0000 |
| 10 | 0.1006 | 0.4407 | 5.8773 | 0.0000 | 0.0000 |
| 11 | 0.1572 | 6.2896 | 1.1341 | 0.0059 | 0.0000 |
| 12 | 0.0000 | 0.6583 | 2.1129 | 0.7137 | 0.0000 |
| 13 | 0.2264 | 0.3482 | 1.1341 | 1.0750 | 0.0000 |
| 14 | 0.3081 | 1.0664 | 2.1129 | 0.2123 | 0.0000 |
| 15 | 1.8172 | 1.5724 | 0.7577 | 0.1475 | 0.0010 |
| 16 | 0.0252 | 1.7628 | 0.3930 | 0.8494 | 0.0361 |
| 17 | 0.1006 | 0.0218 | 0.7577 | 1.1561 | 0.1213 |
| 18 | 0.7608 | 3.4005 | 1.4659 | 0.0059 | 0.0000 |
| 19 | 0.0566 | 0.0871 | 5.8773 | 0.0531 | 0.0000 |

**Step 5:** Table 22 displays the ideal alternative chosen from Table 18.

**Table 22.** Table of ideal issue.

| | max $W_{i1}$ | max $W_{i2}$ | max $W_{i3}$ | max $W_{i4}$ | max $W_{i5}$ |
|---|---|---|---|---|---|
| S+ | 1.8172 | 6.2896 | 5.8819 | 1.1561 | 0.1213 |

**Step 6:** Table 23 displays the nadir alternative chosen from Table 18.

**Table 23.** Table of nadir issue.

| | min $W_{i1}$ | min $W_{i2}$ | min $W_{i3}$ | min $W_{i4}$ | min $W_{i5}$ |
|---|---|---|---|---|---|
| S− | 0.0000 | 0.0218 | 0.3976 | 0.0000 | 0.0000 |

**Step 7:** By using the Equations (4) and (5), a distance measure over each criterion to both ideal ($D^+$) and nadir ($D^-$) is developed. The same is shown in Table 24.

**Table 24.** Distance of ideal and nadir issue from weighted data.

| Issue | $D_i^-$ | $D_i^-$ |
|---|---|---|
| Initial cost of the MH equipment | 10.9437 | 1.7833 |
| Load carrying capacity | 10.4622 | 2.6358 |
| Programming flexibility of MH equipment | 10.3733 | 2.7948 |
| Operational cost | 9.5797 | 3.4227 |
| Throughput rate | 10.8597 | 2.0683 |
| Capacity to handle different shapes and volumes | 10.3192 | 2.8975 |
| Storage/Retrieval MH equipment | 9.7465 | 3.2452 |
| Operational control | 9.7077 | 4.3243 |
| Automation | 9.6648 | 3.2041 |
| Floor space | 9.6956 | 5.8947 |
| AGVs/Robots and other advanced MH equipment already present | 9.0706 | 6.3930 |
| Number of AGVs required | 10.4161 | 2.3253 |
| Layout of AGV tracks | 10.8966 | 1.6169 |
| Vehicle dispatching rules | 10.3076 | 2.3961 |
| Traffic management | 10.4425 | 2.5240 |
| Positioning of idle vehicles | 10.6712 | 1.9964 |
| Failure management | 11.3093 | 1.3914 |
| Compatibility of MH equipment with other workstations | 9.4341 | 3.7804 |
| Comparison with cheap human labour | 9.9357 | 5.8785 |

**Step 8:** A ratio R is determined for each issue which is equal to the distance to the nadir divided by the sum of the distance to the nadir and the distance to the ideal, as shown in Equation (6) and calculated in Table 25.

**Step 9:** Rank the issues by maximizing the ratio in Step 8 as shown in Table 26.

**Table 25.** Ratio of distance to nadir from total.

| Issue | $R_i$ |
|---|---|
| Initial cost of the MH equipment | 0.1401 |
| Load carrying capacity | 0.2012 |
| Programming flexibility of MH equipment | 0.2122 |
| Operational cost | 0.2632 |
| Throughput rate | 0.1600 |
| Capacity to handle different shapes and volumes | 0.2192 |
| Storage/Retrieval MH equipment | 0.2498 |
| Operational control | 0.3082 |
| Automation | 0.2490 |
| Floor space | 0.3781 |
| AGVs/Robots and other advanced MH equipment already present | 0.4134 |
| Number of AGVs required | 0.1825 |
| Layout of AGV tracks | 0.1292 |
| Vehicle dispatching rules | 0.1886 |
| Traffic management | 0.1947 |
| Positioning of idle vehicles | 0.1576 |
| Failure management | 0.1096 |
| Compatibility of MH equipment with other workstations | 0.2861 |
| Comparison with cheap human labour | 0.3717 |

**Table 26.** Ranking the issues from largest to smallest value.

| Issue | $R_i$ |
|---|---|
| AGVs/Robots and other advanced MH equipment already present | 0.4134 |
| Floor space | 0.3781 |
| Comparison with cheap human labour | 0.3717 |
| Operational control | 0.3082 |
| Compatibility of MH equipment with other workstations | 0.2861 |
| Operational cost | 0.2632 |
| Storage/Retrieval MH equipment | 0.2498 |
| Automation | 0.2490 |
| Capacity to handle different shapes and volumes | 0.2192 |
| Programming flexibility of MH equipment | 0.2122 |
| Load carrying capacity | 0.2012 |
| Traffic management | 0.1947 |
| Vehicle dispatching rules | 0.1886 |
| Number of AGVs required | 0.1825 |
| Throughput rate | 0.1600 |
| Positioning of idle vehicles | 0.1576 |
| Initial cost of the MH equipment | 0.1401 |
| Layout of AGV tracks | 0.1292 |
| Failure management | 0.1096 |

## 7. Discussion

The aim of this research paper is to identify the different material handling issues in an advanced manufacturing system like FMS. For this 19 issues have been identified which influence the MH system of FMS. Further these are modelled using two distinct, well established modelling approaches, ISM and TOPSIS. In ISM based model a hierarchy of different issues based on their relative importance is established. The practising managers of these industries can get an insight of these issues and understand their relative importance and interdependencies. The drive power- dependence power (Table 19) gives some valuable insights about the relative importance and interdependence among these issues. Research indicates that positioning of idle vehicles, failure management, traffic management, throughput rate, layout of AGV tracks and vehicle dispatching rules and so forth are the top level issues. They have less influence and more dependence on the other MH issues. Initial cost, number of AGVs, compatibility of different MH equipment with other processing workstations and AS/RS devices, operational cost, operational control, automation, load carrying capacity and the programming flexibility form the middle level attributes. They have both the driving as well as dependence power. So they influence as well as are influenced by the other MH issues. The capacity to handle different shapes and volumes, storage and retrieval MH equipment, comparison with cheap human labour, floor space and MH equipment already present form the lowest level issues. The ISM model suggests the lowest level issues have a very high driving power and as such influence all other issues.

This implies for the proper design of a MH system in FMS the main criteria/issue is the floor space and the existing MH system. Further, comparison with cheap human labour is also important. Especially in countries, where cheap labour is available, it may not be economically viable option to select and use the sophisticated MH equipment. So, some of the researchers have also come up with the concept of a humanised FMS where the material handling tasks in the FMS environment can be carried out by human labour (Nagar and Raj 2012). The other middle level and top level issues although very important for the success of MH system design in FMS can be achieved with the availability of lower level issues.

The ISM model is validated using TOPSIS methodology on the same issues. The hierarchy model given by TOPSIS is similar to that of ISM to a large extent. The TOPSIS also evaluated that the existing MH equipment and the floor space are the most necessary issues to be considered for the design of the MH system, followed by the comparison with cheap labour, operational control and compatibility of

MH equipment with other workstations. So these issues can be treated as the key issues for designing and selecting the MH system for FMS.

## 8. Conclusions

The present work identifies and models the main material handling issues in FMS. The purpose of identification of these issues and their analysis is to allow researchers and practicing managers to pay proper attention to these issues which may help them in designing the material handling systems in their organisations in a better way. The main limitation is this work is that, it is based on the opinions of the experts and industrial personnel which may be biased and based on their individual, personal experiences. This work can be further extended by using some MCDM techniques like AHP to precisely select the equipment required for the MH system.

**Author Contributions:** S.D. has done the planning, design, survey, modelling and writing of the present research. T.R. has the supervision during the research.

**Funding:** This research received no external funding.

**Conflicts of Interest:** The authors declare no conflicts of interest

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
