# Peer review of "A Hybrid MADM Approach for the Evaluation of Different Material Handling Issues in Flexible Manufacturing Systems"

_admsci, doi:10.3390/admsci8040069_

Round 1
Reviewer 1 Report
The paper discusses evaluation of different material handling issues in FMS. The paper introduces the problem and solutions in a detailed matter; however the main source of the whole study is hidden in a different article [1]. This is really a pity setup as the reader cannot see the real numbers/facts that the whole article builds on. I actually would re-write the title as the source for the FMS survey is limited to India, which in default should not reflect the whole worlds’ setup. Through the whole article, I felt a lack of main contribution, what is the main point this paper would like to emphasize?
The authors should argue and describe in details, what is the difference between [1] and the current article.
[1] Dixit, S. and Raj, T. (2018) ‘Feasibility analysis of FMS in small and medium scale Indian 358 industries with a hybrid approach using ISM and TOPSIS’, International Journal of Advanced Operations Management, Vol. 10, No.3, pp.252 – 280.
Author Response
Kindly find the file attached.

Reviewer 2 Report
This paper using a hybrid Interpretive structural modelling (ISM) and Technique for order preference by similarity to the ideal solution (TOPSIS) approach to evaluate the different materials handling issues in Flexible Manufacturing Systems (FMS). The structure of the paper is clear and easy to follow.
suggested changes as below:
reference should be numbered.
there are quite few abbrivations used i.e ISM and TOPSIS. please give a full name before introduce the abbrivations.
in the introduction session, a diagram to illustrate the relations between flexible manufacturing systems and the material handling systems would be useful.
the 19 major issues that the authors listed, are they in rank order or just random being presented?
Section 2, pg 2, the authors claim that "It has been found that 35 to 40% of the plant accidents are due to bad methods of material handling. is there any reference associated with it?
perhaps the authors need to address clearly what is difference with the methods that they proposed (ISM +TOPSIS) and other research proposed.
pg 6 in Table 2, there are 18 issues listed not 19 as authors state.
pg 6 please explain why binary digits 0 and 1 are used in place of symbols V, A, X, O of SSIM to develop the reachability matrix.
please state why it is important to present each iterations ( Table 5 to Table 17). These tables need to be explained in detail. they are the critical information that would help the reader to understand the selection process.
table 19 described in pg11, but presented in pg 14. it should close to each other.
pg 12, Figure1, what is the starting point, what is the end point. same question for Figure 2 as well.
Author Response
Kindly find the file attached.

Round 2
Reviewer 1 Report
The authors have
edited the paper according the reviewer’s comments, all mayor questions are answered.
Author Response
The authors have edited the paper according the reviewer’s comments, all major questions are answered.

Reviewer 2 Report
The authors has made improvement based on the reviewers comments.
There are few comments that the authors just response as "done" and not refer to where in the text, perhaps this need to be more clear.
Author Response

(The authors gave the same response as above.)
